

# Learning about the meanings of ambiguous words: evidence from a word-meaning priming paradigm with short narratives

Lena M. Blott[1], Oliver Hartopp[2], Kate Nation[2] and Jennifer M. Rodd[1]

[1] Division of Psychology and Language Sciences, University College London, University of London, London, United Kingdom

[2] Department of Experimental Psychology, University of Oxford, Oxford, United Kingdom

## ABSTRACT

Fluent language comprehension requires people to rapidly activate and integrate context-appropriate word meanings. This process is challenging for meanings of ambiguous words that are comparatively lower in frequency (*e.g.*, the "bird" meaning of "crane"). Priming experiments have shown that recent experience makes such subordinate (less frequent) word meanings more readily available at the next encounter. These experiments used lists of unconnected sentences in which each ambiguity was disambiguated locally by neighbouring words. In natural language, however, disambiguation may occur via more distant contextual cues, embedded in longer, connected communicative contexts. In the present experiment, participants ($N = 51$) listened to 3-sentence narratives that ended in an ambiguous prime. Cues to disambiguation were relatively distant from the prime; the first sentence of each narrative established a situational context congruent with the subordinate meaning of the prime, but the remainder of the narrative did not provide disambiguating information. Following a short delay, primed subordinate meanings were more readily available (compared with an unprimed control), as assessed by responses in a word association task related to the primed meaning. This work confirms that listeners reliably disambiguate spoken ambiguous words on the basis of cues from wider narrative contexts, and that they retain information about the outcome of these disambiguation processes to inform subsequent encounters of the same word form.

Corresponding author
Lena M. Blott, lena.blott.12@ucl.ac.uk

## INTRODUCTION

Language is highly ambiguous. As over 80% of common English words have multiple dictionary definitions (*Rodd, Gaskell & Marslen-Wilson, 2002*), the ability to rapidly select context-appropriate word meanings is a key component of skilled comprehension. Disambiguation processes are particularly complex for subordinate (less frequently used) word meanings, *e.g.*, when the word "crane" refers to a bird rather than to a tall metal structure on a building site. People spend longer reading ambiguous words that are disambiguated towards their subordinate meaning, compared with unambiguous words
and dominant meanings (*e.g.*, *Duffy, Morris & Rayner, 1988*; *Pacht & Rayner, 1993*). In spoken language, a similar increase in processing time and resources for ambiguous compared to unambiguous words has been shown in studies using a range of approaches including neuroimaging (*e.g.*, *Rodd, Davis & Johnsrude, 2005*; see *Vitello, 2014* for review), dual-task methods (*Rodd, Johnsrude & Davis, 2010*), picture selection tasks (*Foss, Bever & Silver, 1968*) and pupillometry (*Kadem et al., 2020*).

Recent word-meaning priming experiments suggest that learning processes play a key role in word-meaning disambiguation. Following successful disambiguation of an ambiguous word, people retain information about the encounter. The availability of the selected meaning is temporarily boosted for easier access at subsequent encounters of the same word form (*Betts et al., 2018*; *Gaskell, Cairney & Rodd, 2019*; *Gilbert et al., 2018*; *Gilbert et al., 2021*; *Rodd, Johnsrude & Davis, 2010*; *Rodd et al., 2013*; *Rodd et al., 2016*; see *Rodd, 2020* for review). This temporary availability boost represents a form of lexical learning that aids language interactions by supporting access to relatively rare word meanings that are used consistently within texts or conversations. Similarly, work on text reading has found that shifting between the different meanings of an ambiguous word is associated with processing costs (*i.e.*, longer reading times, *Binder & Morris, 1995*; *Binder & Morris, 2011*). More broadly, these findings demonstrate people's sensitivity to their current linguistic environment and the flexibility of the adult lexical processing system (*Gaskell, Cairney & Rodd, 2019*).

The word-meaning priming effect is well replicated. Experiments typically consist of a priming phase in which ambiguous words are strongly disambiguated by their sentence context followed by a test phase. Tasks such as word association (*Betts et al., 2018*; *Gaskell, Cairney & Rodd, 2019*; *Gilbert et al., 2018*; *Gilbert et al., 2021*; *Rodd et al., 2013*; *Rodd et al., 2016*) or speeded semantic relatedness judgement (*Gilbert et al., 2018*; *Gilbert et al., 2021*) are used to tap word meaning availability at test. One limitation of these experiments is that the priming phase typically presents the ambiguous words within lists of unconnected sentences that contain strong local disambiguation cues (although see *Betts et al., 2018*; *Rodd et al., 2016* for experiments with longer paragraphs). In natural language, by contrast, disambiguation may also occur based on more distant contextual cues embedded in a longer communicative context (but see *Witzel & Forster, 2015*; *Rice et al., 2019* for discussion of the importance of local lexical cues in disambiguation). The effect of global (rather than sentence-internal, local) context has previously been investigated during reading (*e.g.*, *Binder, 2003*; *Kambe, Rayner & Duffy, 2001*). Irrespective of whether context occurred locally or globally, reading times were longer for ambiguous words when the context was consistent with the word's subordinate (rather than dominant) meaning. The finding that this subordinate-bias effect (*Pacht & Rayner, 1993*) is similar across both local and global contexts shows that readers are able to use more distant contextual cues to disambiguate word meanings further downstream.

In the present work, we explored listeners' ability to process and learn from short narratives where disambiguation was driven by global rather than local context (see examples in Table 1). The priming phase comprised three-sentence narratives that ended with an ambiguous word prime (*e.g.*, ''crane''). The first sentence of each narrative
**Table 1  Example stimuli.**

| | Priming phase | | Test phase |
|---|---|---|---|
| | **Narrative (spoken)** | **Semantic relatedness probe (written)** | **Word association probe (spoken)** |
| Example | Zoe's parents took her to the pet shop to finally choose her birthday gift. When she walked through the door, she was a bit nervous. She could already see the litter. | SURPRISE | litter |

established a situational context congruent with the subordinate meaning of the prime, but the remainder of the narrative provided no disambiguating information. During this priming phase, participants completed a semantic relatedness task in which they responded to visually presented word probes to ensure they were listening for meaning. After a brief delay filled by a digit span task lexical learning was assessed in the form of a word association task. This test task probed the impact of the contextual exposure to the ambiguous word during the priming phase on a subsequent, context-free encounter (*i.e.*, word association task) of the same word forms (*Gilbert et al., 2018*; *Twilley et al., 1994*).

In addition to lexical learning, the word-meaning priming paradigm allowed us to assess the success of disambiguation processes: only if participants have correctly selected the subordinate meaning during the priming phase can the availability of this specific meaning receive a boost. While eye-movement studies have provided evidence that people can use distant contextual cues for disambiguation when reading (*e.g.*, *Hess, Foss & Carroll, 1995*; *Kambe, Rayner & Duffy, 2001*; *Schwanenflugel & White, 1991*), it has proven more difficult to measure equivalent disambiguation processes for spoken language (see *Rodd, Johnsrude & Davis, 2010*). Our paradigm provides an indirect approach to measuring disambiguation in spoken language, and one that has clear advantages over tasks that measure comprehension more directly *e.g.*, via explicit comprehension questions, or by asking people to make semantic relatedness judgements (*e.g.*, CRANE—BIRD) that may themselves change behaviour by explicitly priming particular meanings. Using a priming approach with spoken language, *Gilbert et al. (2021)* have previously replicated key findings from the reading literature. In the priming phase of their experiment, disambiguating information congruent with the subordinate meaning occurred either before or after the ambiguous word. Priming was reduced when disambiguation cues occurred after the ambiguous word, suggesting that listeners were less likely to successfully resolve the ambiguity in late-disambiguation compared to early-disambiguation sentences, in line with findings from the literature on sentence reading (*e.g.*, *Frazier & Rayner, 1990*). Priming was also reduced when the priming task could be completed via relatively shallow processing (listening to sentences with only occasional comprehension questions vs. making a semantic relatedness judgement after every sentence). This is in line with the "good enough" account of sentence processing (*e.g.*, *Ferreira & Patson, 2007*), whereby readers tend to process complex language in a way that is "good enough" for the task at hand - for example, by not always resolving syntactic (*Christianson et al., 2001*) or semantic ambiguities (*Blott et al., 2021*). In their word-meaning priming study, *Gilbert et al. (2021)* found that, like readers, listeners do not always fully engage in disambiguation processes. The presence of a

priming effect in the current experiment will therefore provide insight into the likelihood of disambiguation success when *spoken* contextual cues to disambiguation are relatively distant from the ambiguous word itself, potentially replicating findings from the literature on written sentence processing (*e.g.*, *Hess, Foss & Carroll, 1995*; *Kambe, Rayner & Duffy, 2001*; *Schwanenflugel & White, 1991*)

   We hypothesised that subordinate meanings would receive an availability boost when they had recently been encountered (Primed condition) compared to when they had not (Unprimed condition, see preregistration). To preface an exploratory analysis without a specific hypothesis, we also investigated whether priming is modulated by word meaning dominance. *Rodd et al.* (*2013*, Exp. 1) found greater priming for subordinate meanings of words that were strongly biased (*e.g.*, critter vs listening device "bug") compared to words whose meanings are relatively equal in frequency (*e.g.*, mammal vs container "seal"). Plausibly, this finding reflects the fact that a single encounter with a meaning is highly informative when the described event is unexpected—as is the case for more strongly subordinate meanings. However, it is also plausible that strongly subordinate meanings might be less reliably disambiguated, resulting in reduced priming for these items. Any effect of dominance on word-meaning priming (*i.e.*, an interaction of Dominance × Priming) in the present data may help to understand dominance within theoretical accounts of word-meaning priming (*e.g.*, by taking into account baseline dominance as a limit to possible shifts in meaning availability through priming, *Rodd, 2020*), and guide stimulus selection in future studies.

   In summary, the current experiment assessed whether participants can disambiguate spoken ambiguities on the basis of distant contextual cues, whether these disambiguation encounters are used for lexical learning and affect word meaning interpretation at a later time point, and to what extent such priming effects are modulated by word meaning dominance.

## MATERIALS & METHODS

Hypotheses, methods and data analysis plan were preregistered, with deviations noted where necessary: https://osf.io/xwn5h. Data and analysis code are available here: https://osf.io/bvnhk/.

### Participants

We recruited 60 participants via Prolific (https://www.prolific.co/, *Damer & Bradley, 2014*). Based on recommendations by *Brysbaert & Stevens (2018)*, we aimed for a sample size that would give at least 80% power to detect a priming effect. Sixty participants would provide 1,980 datapoints per condition, exceeding *Brysbaert & Stevens (2018)* recommendation for 1,600 observations per condition for a mixed effects analysis. Prolific's screening tool recruited native English speakers aged 18–40 without language, literacy or hearing impairments and with Prolific approval ratings >75%. Following preregistered criteria, six participants were excluded due to uncorrected visual/hearing impairment ($n = 2$), a language disorder ($n = 1$), failing to follow instructions (*e.g.*, by repeating back the word association probes, $n = 3$), or not completing the word association task ($n = 3$). Analyses

were based on data from 51 participants ($M_{age}$ = 29 years, range 18–40). Note that we had intended to exclude participants with less than 100% accuracy on a Huggins Pitch task to ensure that they were wearing headphones (see preregistration). Although 12 participants failed this task, a programming error meant they were not automatically excluded. However, their performance on the auditory tasks (semantic relatedness, $Min_{accuracy}$ = 80%, and responses given on the word association task) indicated they could hear the materials sufficiently well. We therefore deviated from the preregistration and included their data.

Participants provided informed written consent via a web-form, and were remunerated at £10/hour. The study was approved by the by the Medical Sciences Interdivisional Research Ethics Committee at the University of Oxford (approval number: R73564/RE001).

## Procedure

The experiment was programmed using Gorilla experiment builder (http://www.gorilla. sc/about; Cauldron Inc.; *Anwyl-Irvine et al., 2019*), and completed online. Participants used their own computer (the experiment was not accessible via a phone/tablet) and were instructed to wear headphones. The experiment comprised six phases (Fig. 1), which were not separated by enforced breaks between them—instead, participants automatically moved from one task to the next.

Instructions to enable the browser "autoplay sounds" option were provided in the audio-check phase (*Milne et al., 2021*). Participants were reminded to wear headphones before completing six trials of a Huggins Pitch task, which includes tones that are only audible through headphones (*Milne et al., 2021*). Participants then listened to an example sentence and adjusted computer volume levels accordingly.

In the priming phase, each participant listened to 33 narratives that ended in an ambiguous prime, and 33 unambiguous fillers (items were counterbalanced across participants, see Materials for details). Trial order was randomised for each participant. To encourage listening for meaning, they made simple semantic relatedness decisions to visually presented probe words at the offset of each spoken narrative (responding 'c' for related, or 'm' for unrelated probes). In total, 50% of trials were followed by a related probe. Responses did not time out. The experiment began with four practice trials (three unambiguous fillers and one ambiguous narrative similar to the experimental stimuli). Participants received accuracy feedback for practice trials only. The main task started with four additional unambiguous filler narratives to allow participants to become accustomed to the trial structure. Attention was not drawn to the presence of ambiguous words and participants were not told that some words would appear again later in the test phase.

A forward and backward digit span task from the Gorilla Open Materials depository (*Dean, 2020*) provided a delay between the priming and the test phases ($M = 6.86$ min, SD = 1.86, Range = 4–13 min).

In the test phase all 66 ambiguous words were presented aurally in a randomised order for word association. Participants were instructed to type the first word that came to mind. Each trial timed out after 30 s. The mean difference between the estimated average mid-point of the priming phase and the mid-point of the test phase was approximately 20

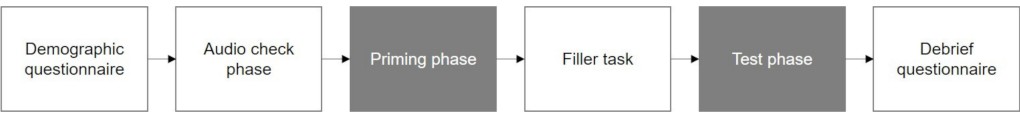

**Figure 1    Illustration of the experimental procedure.**

min, which is in line with previous studies on word meaning priming (*Betts et al., 2018*; *Gilbert et al., 2021*; *Rodd et al., 2013*).

Finally, participants were asked two open-ended questions in the debrief questionnaire: "Did you notice anything about the stories or words that were presented to you?" and "What do you think the aim of this study was?".

## Design

We used a repeated-measures design with the single factor Priming. Participants were randomly allocated to an experimental version ($N = 27, 24$). Each participant was primed with half the ambiguous words, but tested on all ambiguous words. Across versions, each ambiguous word appeared in both the Primed and the Unprimed condition. The dependent variable was whether each word association response was consistent with the subordinate meaning of the ambiguous word.

## Materials

Prime words were 66 noun-noun homophones (Table 2) taken from a wider stimulus pool (*Blott et al. 2022*, preprint). We identified subordinate meanings based on spoken word association norms (*Gilbert & Rodd, 2022*); 68% of primes were homographs (where meanings share orthographic form and pronunciation). We chose ambiguous words that were associated with only two meanings, or had two very clear major meanings with any other meanings being extremely infrequent or niche (*e.g.*, the norms indicated that, apart from "limbs" and "weapons", the ambiguous word "arms" had a niche third meaning: the "name of a pub"). This choice was made to ensure that our results would not be affected by the number of potential meanings of our ambiguous primes (*Yip, 2015*). Measures of word length, frequency, familiarity and age of acquisition in Table 2 came from the LexOps shiny app (*Taylor, Beith & Sereno, 2020*). In lieu of more formal measures of the primes' predictability or surprisal within the narrative contexts, we also report a measure of semantic similarity between the primes and the remainder of their associated narrative, based on latent semantic analysis (LSA) using http://lsa.colorado.edu/ based on *Dennis (2007)*. Semantic similarity scores can range from −1 (low) to +1 (high). Here, scores had a mean of .06, suggesting that the narratives did not tend to contain strong lexical associates with the primes. While this is a crude measure, it indicates that the final words in both ambiguous and unambiguous narratives were relatively unpredictable. We did not use LSA to generate our stimuli.

We constructed 66 three-sentence narratives that ended with an ambiguous prime (Table 1). Sentence 1 provided a context in which the subordinate meaning of the prime was more plausible than the dominant meaning. Sentence 2 introduced a short delay following the

**Table 2   Descriptive statistics of the primes.**

| Measure | Mean (SD) | Range |
|---|---|---|
| Length (number of syllables, calculated by eSpeak speech synthesiser for British pronunciations; http://espeak.sourceforge.net/) | 1.43 (0.59) | 1–3 |
| Frequency per million words (SUBTLEX-UK, *Van Heuven et al., 2014*) | 44.91(80.46) | 0.39–489.02 |
| Age of acquisition (Glasgow norms, *Scott et al., 2019*) | 3.14 (0.85) | 1.66-5.25 |
| Familiarity (Glasgow norms, *Scott et al., 2019*) | 5.64 (0.64) | 4.18-6.71 |
| Meaning dominance (proportion of word association responses consistent with the dominant meaning, based on *Gilbert & Rodd, 2022*) | 0.78 (0.14) | 0.51-0.99 |
| Semantic similarity between primes and narrative (based on latent semantic analysis, http://lsa.colorado.edu/; ranging from −1 (low) to +1 (high) similarity in co-occurrence statistics) | 0.06 (0.12) | −0.11-0.5 |

disambiguating information, and was uninformative with respect to the meaning of the prime. Sentence 3 ended with the prime but on its own, this sentence could plausibly be interpreted using either meaning of the prime. The mean distance between Sentence 1 and the prime contained in Sentence 3 was approximately 12 words (range: 7–23 words); it was not possible to tightly control distance across the stimulus set as the location of the cues within Sentence 1 and the length of Sentences 2 and 3 varied.

For each narrative, we selected a single word to be presented visually as a probe for the semantic relatedness task during the priming phase (see Table 1 for an example). Related probes were associated with the global content of the narrative, not the meaning of prime words. Unrelated probes were unrelated to both the overall narrative and the meanings of the prime. Each narrative was paired with the same relatedness probe for all participants.

Thirty-three filler narratives served to reduce the salience of the ambiguous primes. They were constructed around a set of unambiguous filler words in narrative-final position.

Narratives and word association probes were presented aurally. They were recorded by a male speaker of Southern British English in a sound-attenuated booth (sampling rate 44.1 kHz) and processed in Praat (v 6.1.16, *Boersma & Weenink, 2020*). A silent period of ~200 ms preceded the onset of speech; each sound file ended at speech offset. Sound files were down-sampled to 22,050 kHz and intensity scaled to 60dB. The files were band-pass filtered from 60–20,000 Hz (smoothing factor 10) and converted into stereo .mp3 files using Audacity v2.3.3.

The 66 narratives were split into two lists with a similar distribution of relative meaning dominance values for use in the two experimental versions. In both lists, 16 of the ambiguous narratives were paired with a related probe, 17 with an unrelated probe. Each stimulus list also contained the 33 filler narratives (17 paired with related probe, 16 paired with unrelated probe).

## RESULTS

### Priming phase: semantic relatedness task

Mean accuracy for ambiguous trials was high ($M = 0.95$, SD $= 0.22$), and no participant scored below 70%, confirming that participants were listening for meaning.

### Test phase: word association task

Each response was coded as either consistent with the subordinate meaning used in the context of the priming phase (coded as '1') or inconsistent (coded as '0'). Responses were independently coded by two authors, blind to priming condition. We excluded responses that were related to both meanings or likely reflected mishearings or were otherwise unclear. The coders initially agreed on 97.3% of decisions. Disagreements were resolved by discussion. 332 trials (8.98%) were excluded.

As per the preregistration, the data were analysed with logit mixed effects models (*Baayen, Davidson & Bates, 2008*; *Barr et al., 2013*), using the glmer() function (with bobyqa optimiser) within the lme4 package (v1.1.26; *Bates et al., 2015*) in RStudio (v3.6.2; *RStudio Team, 2015*). The factor "Priming" was deviation-coded (Primed 0.5, Unprimed $-0.5$). Our preregistration specified that we would include a fixed effect for stimulus list; however, the two factors of Priming and List were redundant, so we excluded the fixed effect for List. Although not specified in the preregistration, our maximal model also included random slopes for Priming to account for inter-individual and inter-item differences in the effect of priming on word association responses. Due to convergence issues, we reduced the complexity of the random effects structure (following *Barr et al., 2013*); the maximal model included a fixed effect for Priming and random intercepts by participants and items.

As predicted, responses were more likely to be consistent with the subordinate meaning if participants had encountered that meaning in the Priming Phase (Fig. 2; $b = 0.53$, SE $= 0.10$, $z = 5.13$; $\chi^2(1) = 25.77$, $p < .001$).

For an exploratory analysis of how a word's relative meaning dominance affected priming, we used a measure of dominance previously calculated from an independent data set of word association responses (*Gilbert & Rodd, 2022*). This measure captures the likelihood with which people produce responses associated with the meanings of an ambiguous word, as a value between 0 (both meanings equally likely) and 1 (one meaning produced exclusively). We fitted a model with a continuous fixed effect for the item's dominance (D as calculated in *Gilbert & Rodd, 2022*), which was centred before inclusion in the model, a fixed effect of Priming and their interaction. Due to convergence issues, we reduced the complexity of the random effects structure following recommendations in *Barr et al. (2013)*. The maximal model that converged included random intercepts by participants and by items. This analysis found a significant main effect of Dominance; compared to items with lower Dominance values, items with higher baseline Dominance were associated with fewer responses consistent with the subordinate meaning ($b = -5.63$, SE $= 0.58$, $z = -9.77$; $\chi^2(1) = 66.1$, $p < .001$). This negative relationship between baseline dominance and consistency of word association responses is illustrated in Fig. 3. It confirms that the participants in this experiment had similar meaning preferences to those sampled by *Gilbert & Rodd (2022)*. However, in contrast with our prediction, Dominance did not

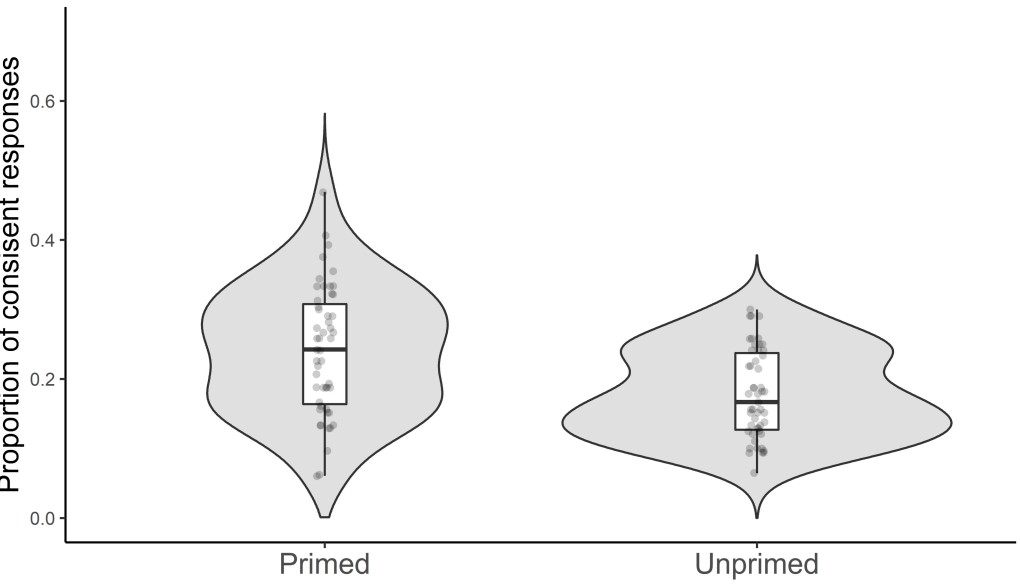

**Figure 2** **Mean proportion of consistent responses in the word association task.** Each dot represents a participant and shows the proportion of responses consistent with the intended (subordinate) meaning of the ambiguous words as a function of priming condition. The boxplots represent median and quartiles across all participants in each condition.

interact with Priming (see Fig. 4); there was no evidence to suggest that the magnitude of priming was dependent on the baseline dominance of the item ($b = 0.59$, SE $= 0.44$, $z = 1.34$; $\chi^2(1) = 1.75$, $p = .186$). Importantly, the main effect of Priming on consistency remained significant when baseline dominance was included in the model ($b = 0.63$, SE $= 0.13$, $z = 5.07$; $\chi^2(1) = 25.41$, $p < .001$).

### Debrief

In response to the question "Did you notice anything about the stories or words that were presented to you?" 8/51 participants (15.69%) reported they had noticed words from the semantic relatedness task being repeated later in the experiment, and 11/51 participants (21.57%) noticed that some words had multiple meanings. We did not explore this further as the present experiment was not powered to detect effects of awareness, but have made the data freely available on the OSF, should other researchers wish to pursue questions about the effects of participants' awareness of our experimental manipulation.

## DISCUSSION

This study found that a single encounter with the subordinate meaning of an ambiguous word within a spoken narrative increased the availability of this meaning at a subsequent encounter, replicating the word-meaning priming effect (*Betts et al., 2018*; *Gaskell, Cairney & Rodd, 2019*; *Gilbert et al., 2018*; *Gilbert et al., 2021*; *Rodd et al., 2013*; *Rodd et al., 2016*). Notably, our experiment differed from previous work as ambiguous words occurred within narratives rather than in a list of unconnected sentences, and disambiguation was driven

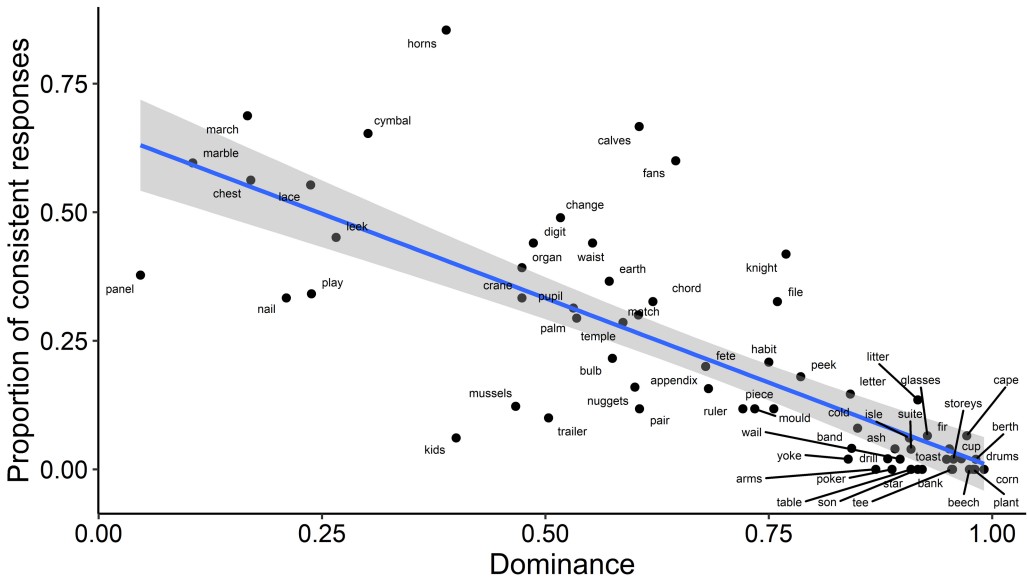

**Figure 3  Relationship between mean consistency of responses with the subordinate meaning and baseline dominance of the ambiguous words.** A dominance score of 0 describes a word whose meanings are entirely balanced in terms of the frequency with which they were produced by respondents in word association tasks (*Gilbert & Rodd, 2022*), and 1 describes a word for which word association responses were biased towards a single, *i.e.*, dominant, meaning. Consistency of responses with the subordinate meaning was averaged across the Primed and Unprimed condition to illustrate the main effect of Dominance on response consistency.

by contextual cues separated from the ambiguous word by an intervening sentence, rather than by sentence-internal lexical cues. Participants showed evidence of lexical learning in the present experiment. That is, they retained information about the meanings they had encountered, and this guided their subsequent interpretation of the words. We provide evidence that lexical learning can occur through naturalistic comprehension processes, even when successful comprehension relies on the use of global contextual cues to disambiguate strongly subordinate word meanings. The present work adds to the evidence that word-meaning priming is a robust phenomenon (*Betts et al., 2018*; *Gilbert et al., 2021*; *Rodd et al., 2013*; *Rodd et al., 2016*). Indeed, priming effects have now been observed when the priming phase required disambiguation based on local sentence-internal disambiguation cues occurring before (*Rodd et al., 2013*; *Rodd et al., 2016*) or after the ambiguous word itself (*Gilbert et al., 2021*), and based on distant disambiguation cues from earlier in a paragraph (*Betts et al., 2018*) or narrative (present study).

Although the availability of subordinate meanings was boosted by priming, the baseline dominant meanings were still preferred in the word association task overall. This is a consistent finding in the word-meaning priming literature, with the average proportion of word association responses consistent with subordinate meanings usually not exceeding about 0.35 (*Gilbert et al., 2018*; *Gilbert et al., 2021*; *Rodd et al., 2013*; *Rodd et al., 2016*), and even when the subordinate meaning was primed multiple times or prime encounters were spaced across time (*Betts et al., 2018*). This suggests that prolonged and consistent

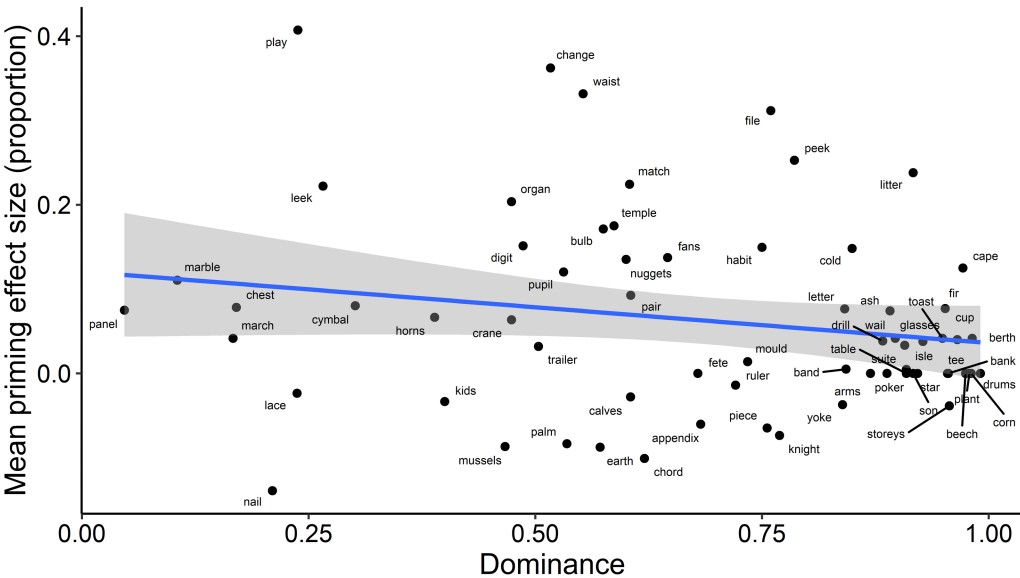

**Figure 4** **Relationship between priming effect size and baseline dominance of the ambiguous words.** A dominance score of 0 describes a word whose meanings are entirely balanced in terms of the frequency with which they were produced by respondents in word association tasks (*Gilbert & Rodd, 2022*), and 1 describes a word for which word association responses were biased towards a single, *i.e.*, dominant, meaning. Priming effect size in the present experiment was calculated for each item individually by subtracting mean consistency of word association responses with the subordinate meaning in the Primed condition from mean consistency of word association responses with the subordinate meaning in the Unprimed condition.

changes in the linguistic environment may be needed to affect larger changes to the baseline meaning availability that has accumulated through long-term experience across the lifespan (cf *Nation, 2017*). This type of change might be induced by a change in an individual's hobbies or profession, for example (*Eligio & Kaschak, 2020*; *Rodd et al., 2016*; *Wiley, George & Rayner, 2018*).

Baseline dominance had a clear effect on the availability of the subordinate meanings within the experiment: the less strongly an ambiguous word was biased towards a dominant meaning, the more likely participants were to respond with the subordinate meaning. In contrast to *Rodd et al. (2013)* however, and despite including ambiguous words with a relatively wide range of dominance values, there was no evidence to suggest that baseline dominance modulated the size of the priming effect. This null finding should be interpreted with caution but we speculate that for the type of contextual disambiguation required in the current study, the magnitude of priming may be more strongly influenced by other item-specific factors, such as the strength of the disambiguating context (see *e.g.*, *Colbert-Getz & Cook, 2013*; *Kambe, Rayner & Duffy, 2001*; *Martin et al., 1999* for evidence that contextual strength can modulate disambiguation processes during reading). Such factors may have masked any effects of baseline dominance. Further studies that explicitly manipulate word-meaning dominance and contextual strength are needed to disentangle the factors that drive successful disambiguation in spoken language comprehension, and

the subsequent effects of lexical learning from encounters with ambiguous word meanings and how they build up over time.

The nature of the cognitive mechanisms that support the short-term shifts in meanings preferences induced by word-meaning priming is still debated. While the present study cannot answer questions about the underlying mechanisms of these priming effects, it will be important for future work to distinguish between accounts that see the origin of word-meaning priming in an immediate alteration of long-term memory (*Rodd et al., 2016*; *Gilbert et al., 2018*), and those that see the priming phenomenon as the result of a reinstantiation during the testing phase of a gist-like episodic memory representation of the word-meaning encounter during the priming phase (*Curtis et al., 2022*; *Gaskell, Cairney & Rodd, 2019*). Our findings show that successful disambiguation for spoken ambiguities can occur based on contextual representations built earlier in the discourse. This result complements findings from eye-tracking studies in the written modality (*e.g.*, *Hess, Foss & Carroll, 1995*; *Kambe, Rayner & Duffy, 2001*) and extends them to spoken language comprehension, using an implicit measure of disambiguation success.

## CONCLUSIONS

This experiment is an important step towards using more naturalistic and varied forms of disambiguation to study how people use cues from recent experience to update their lexical knowledge in a manner that supports comprehension. It contributes to the growing literature demonstrating the flexibility of the adult lexical processing system (*Rodd, 2020*) and shows the importance of linguistic experience across the lifespan in maintaining high quality lexical representations (*Nation, 2017*).

### Funding
This work was supported by the ESRC (No. ES/S009752/1) and the Department of Experimental Psychology at the University of Oxford. The funders had no role in study design, data collection and analysis, decision to publish, or preparation of the manuscript.

### Grant Disclosures
The following grant information was disclosed by the authors:
ESRC: ES/S009752/1.
Department of Experimental Psychology at the University of Oxford.

### Competing Interests
The authors declare there are no competing interests.

### Author Contributions
- Lena M. Blott conceived and designed the experiments, performed the experiments, analyzed the data, prepared figures and/or tables, authored or reviewed drafts of the article, and approved the final draft.

- Oliver Hartopp performed the experiments, authored or reviewed drafts of the article, and approved the final draft.
- Kate Nation conceived and designed the experiments, authored or reviewed drafts of the article, and approved the final draft.
- Jennifer M. Rodd conceived and designed the experiments, authored or reviewed drafts of the article, and approved the final draft.

### Human Ethics

The following information was supplied relating to ethical approvals (*i.e.*, approving body and any reference numbers):

The study was approved by the by the Medical Sciences Interdivisional Research Ethics Committee at the University of Oxford.

### Data Availability

The data and code are available at the Open Science Framework: Blott, Lena M, Oliver Hartopp, Kate Nation, and Jennifer M Rodd. 2022. ''Word-Meaning Priming with Short Narratives.'' OSF. June 14. https://osf.io/bvnhk/.

### Supplemental Information

Supplemental information for this article can be found online at http://dx.doi.org/10.7717/peerj.14070#supplemental-information.

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
