# Peer review of "Learning about the meanings of ambiguous words: evidence from a word-meaning priming paradigm with short narratives"

_PeerJ, doi:10.7717/peerj.14070_

## Round 0.1 · original submission · Minor Revisions

I have now received two reviews from experts in your field. As you will see from their comments below, both reviewers saw considerable merit in your paper, but both have also made some very constructive suggestions for revision. In particular, Reviewer 1 provides very interesting hints for improving your work. I think it would be important to describe with more precision some aspects of the stimuli and try to contextualize the results obtained with respect to previous literature, providing the information for direct comparison.

At this stage, I am inviting you to submit a revised version of the paper for consideration, carefully addressing each of the reviewers' comments.


Reviewer 1 ·

Basic reporting

The basic reporting in this article is excellent --- the writing is clear throughout, the tables and figures are relevant, the raw data and code is shared, and the hypotheses are all clear and relate to the experiment and corresponding data. My main point of concern with respect to basic reporting is to add some additional text describing the similarity in the main manipulations and results of interest (distance of disambiguating context from ambiguous word, delay between exposure and test) to prior studies on this subject. I elaborate on this point in my detailed comments below.

Experimental design

This article meets all of the journal's standards with respect to experimental design. It is original research with a well-defined and meaningful set of implications for the ambiguity literature, the experiment is conducted in a rigorous way (modulo a few minor concerns I note in my detailed comments below), all relevant ethical standards are met, and the experiment is readily replicable with the supplied materials.

Validity of the findings

The core experiment has been conducted to a high standard and I would expect these results to be replicable using the supplied materials. The statistical tests are all appropriate for the design in question and the overall conclusions are grounded in the empirical findings. I have minor suggestions to improve the article on this front in my detailed comments below.

Additional comments

Major comments:

Disambiguating the theoretical accounts of dominance effects (cf. Rodd, 2020)
At the end of the introduction, around lines 105-108 the authors suggest that the presence of an effect of dominance would help advance theories of word-meaning priming. I am not sure that the paper delivers strong conclusions on this specific front, however. Yes, there is a main effect of dominance in the paper, which does establish that dominance effects are robust across domains and measures. However, the authors also note that their results are inconsistent with Rodd et al. 2013 as they fail to find an interaction between dominance and priming, but that the properties of the stimuli in the current study limit the degree to which this inconsistency can be understood. Perhaps the authors could revise their introduction to make it clearer to readers what specific hypothesis their experiment and stimuli are able is able to test and how that relates to advancing the literature with respect to the aforementioned issue raised in the Rodd (2020) paper?

How often does distal information play a critical role in disambiguation?
The authors claim right from their abstract that "disambiguation *often* occurs via more distant contextual cues" (emphasis mine). There is clear theoretical justification for understanding how distal information shapes processing of ambiguity regardless of how often distal information is of critical importance. However, the importance of this finding would be even greater if distal information truly is critical on a frequent basis and I am not sure that this is true. Based on the findings of Rice et al. (2018) for homonyms of the type studied in the present work, a local sentence context (e.g., a line of dialog from a movie) provides sufficient grounds for disambiguating a homonym in about 90% of sentences. I therefore suggest that the authors operationalize and provide evidence for what they mean by "often," or otherwise temper their language on this aspect of their work. The current work is important and worth reading regardless of the frequency of critical distal information.

Comparison of the impact of distance from disambiguating information and delay in prior work

The two aspects of prior published work that I thought could benefit from some additional treatment in the manuscript is how distance (time?) from the disambiguating information and time between the priming and test phases relate to prior published work, including considerable work by the senior author, but also by others. For example, this is not the first study to examine more distal contexts (cf. the work by Kambe et al. (2001) and Binder (2003) using eye-tracking). Slightly more distinct from the present work, but still thematically related to it, is a body of work focused on how disambiguating two occurrences of the same homonyms does (or does not) cause conflicts between the first and second accessed meanings (e.g., Binder & Morris, 1995; Binder & Morris, 2011; Rayner et al., 1994; Simpson and Kang 1994; Gernsbacher et al., 2001). It also seems that the current study sits somewhere in between several other studies examining the effect of delay between prime and test (e.g., Rodd et al., 2013, 2016). It could be useful to review some concrete numbers from prior published work in the manuscript to better contextualize where this paper falls in terms of the distance between the disambiguating context and the ambiguous word and the delay period, and as a result of these factors what types of effects might be expected in terms of priming, dominance, etc.


Experimental stimuli:
The experimenters tested 66 noun-noun homophones, of which 68% were also homographs. Thus, I assume there were 44 pure homonyms and 22 non-homograph homophones. I wondered the extent to which the inconsistency in orthography-to-phonology mappings could have modulated any of the observed effects by up-regulating competition between the conflicting representations of meaning through an indirect pathway that supports the activation of two different meanigns? Perhaps this could be checked in a basic way by only analyzing each corresponding subset of the data and examining whether the core patterns of significance hold?

Awareness of the manipulation
Similar to my point above about potential heterogeneity in the effects across subsets of the experimental stimuli, I have even greater concerns regarding the potential heterogeneity of the effects across participants, particularly when 15-22% of participants seemed to be roughly aware of the aims of the experiment. I appreciate that the experiment might not have been designed to have power to examine these effects, but some preliminary analysis, potentially using a Bayesian approach that can support or fail to support the null in more explicit terms, could help contextualize the potential relevance of this aspect of the data. The relatively high levels of awareness might also justify including fillers in the test task, which I comment on below, and might negate some of the benefits of using a non-explicit test task.



Other minor comments:


-37: although I appreciate the intuition that disambiguation is more "demanding" for subordinate word meanings, I'm not sure if or how this would align with a mechanistic account of what is happening. For instance, if settling times were longer (as they were for homonyms overall in Rodd et al., 2004) we could replicate the core finding of slower RTs without any assumption about the item being more demanding for the system to process. I defer to the authors' preferences, but I suggest they consider removing this language from the paper since it is not central to their claims.

-77: As far as I remember, the Twilley et al. 1994 paper referenced here only deals with context-free encounters of homonyms. It might be useful to clarify what part of the sentence the Gilbert et al. and Twilley et al. citations relate to.

-90: without referring to the Ferreira & Patson (2007) paper, it is not clear why "shallow, good enough" processing is relevant here. I suggest the authors unpack this idea if it is important to explaining the disambiguation processes described in this paper.

-around line 95, it would potentially be helpful to have an example of the types of stimuli used in the current experiment to make the present work more concrete. Perhaps Table 2 could be referenced earlier?

-around 119: could the authors' comment on what effect size they were targeting and for what final number of participants (i.e., after dropping some participants for various reasons)? This relates to the expected power for the final number of analyzed participants and the strength of inference we should place on some of the null effects.

-161: why only include the ambiguous items and not the fillers? They would not even need to be coded if that is the rate-limiting factor, and this would exclude potential blocking/strategic effects in responding due to only using ambiguous items. I use the word *potential* here because I don't think anyone has ever looked at this factor in prior work. It shouldn't impact the core thrust of the current results either way, but I am curious about the motivation for this decision. Including fillers could also give a basic control for the proportion of responses relevant to the probe and potentially for RTs as well, which could let you determine if homonyms involve more extensive processing overall in this task.

-around 184: The proposal to measure the semantic similarity between the prime and the remainder of the associated narrative is well-founded. However, I am skeptical of using LSA for this purpose anymore as that model was trained on a small corpus of data and has not been found to perform well in relation to other aspects of ambiguity (cf. Beekhuizen et al., 2021). Given the correlations are all near floor and I don't have a strong reason to imagine why strength of association with the narrative would be confounded with other variables of interest, I don't think this necessarily needs to be revised here. However, using more modern models (e.g., word2vec, BERT) would provide a much stronger basis for arguing that the items are all equivalent in terms of their association with their broader narrative.

-around 234: the authors report significant relative effects of priming, but the absolute effects of priming remain relatively small and most responses would still appear to be to the dominant meaning. Could the authors comment on what this means for a theory of ambiguous word priming? These data might suggest that priming can be observed but that the dominant meaning is still very clearly much more active overall despite not being contextually relevant.

-around 239-244. I had to re-read this section a few times to tease apart what effects were attributable to the main effect of dominance, the main effect of priming, and the interaction term. I think my confusion stemmed in part from the order in which the main effects and interaction effects were introduced, as well as the specific wording used here. Perhaps this could be revised for clarity? I also thought that a plot of the effects in the exploratory analysis would have been helpful to visualize this analysis.

-around 267: It was not clear to me how "our findings confirm predictions made by Gilbert and colleagues that word-meaning priming effects are not dependent on the precise cognitive mechanisms involved during disambiguation processes." I think the reader would find it helpful if this point was fleshed out a little more.

-296: it is not clear to me how the present work "shows the importance of linguistic experience across the lifespan in maintaining high quality lexical representations (Nation, 2017)"

-I don't expect this comment to be considered as part of any possible revision, but when I read the section around line 100-107, I had the impression that the authors were considering teasing apart the role of a surprisal account of facilitation for an unexpected (subordinate) meaning versus a more classic priming effect. I could imagine that these two effects could produce distinct predictions --- a very unexpected subordinate could generate very high surprisal and therefore extensive facilitation, whereas one might imagine that priming effects for subordinates might be fairly comparable (so long as clear floor and ceiling effects are avoided). I think this could be a very useful line for future work, and Rastle and colleagues may have recent work that could facilitate an examination of this type of prediction in your own stimuli.

·

Basic reporting

This paper examines an interesting topic on lexical disambiguation process using a word-meaning priming technique. Design is clear and the results are easy to follow. In general, this paper is good and quite well-written. However, the paper may be a bit difficult to general readers and so I suggest the authors can clarify some issues and further supplement some more information to the current version.

Experimental design

The theoretical background and rationale of the study are adequately described and explained but it’s better to explain more on: (1) “distant contextual cues”, how distant between cue(s) and target? (2) Can the paradigm also work to “activate” the other meanings (e.g., the third or fourth meaning) associated with the ambiguous word?

Methodology is clear and appropriate, but more information is necessary. For example, it is more useful to elaborate how to use the LSA to generate the short narratives. Also, in the online experiment, are there any short breaks between each task in the Gorilla setting?

Validity of the findings

Result and discussion are good. One minor query, do you think the numbers of meaning of the ambiguous word affected the size of priming effects when you are analyzing the word meaning dominance?

Additional comments

In addition, this paper is quite well-written, but I found a very few typos in the paper. I suggest that the authors should check that carefully to enhance the readability of the paper.

Anyway, it's a good paper.

Very minor issues:
(1) The reference of “Blott, Gowenlock, Nation & Rodd (2022, preprint) - Line 178” seems missing?
(2) There is also a relevant paper (similar to the logic here) examining similar topic in Cantonese Chinese. - Yip, M. C. W. (2015). Meaning Inhibition and Sentence Processing in Chinese. J Psycholinguist Res (2015) 44:611–621.

---

## Round 0.2 · accepted · Accept

Dear Dr. Lena Blott,

As you will see both reviewers think that the manuscript has improved after revision and all concerns have been addressed. I agree with them and I am happy to inform you that this version of the manuscript is fully acceptable for publication in PeerJ.

Reviewer 1 ·

Basic reporting

No issues.

Experimental design

No issues.

Validity of the findings

No issues.

Additional comments

The authors have revised what was already a very strong manuscript into an even stronger re-submission. I thank the authors for the thoroughness of their responses and revisions, and for encouraging researchers to consider the impact of more distal context. I recommend acceptance as is.

·

Basic reporting

No comment

Experimental design

No comment

Validity of the findings

No comment

Additional comments

Most of the concerns were addressed by this revised version. Therefore, I recommend the current version is acceptable for publication.